# Pathogenicity Detection and Genome Analysis of Two Different Geographic Strains of BmNPV

**DOI:** 10.3390/insects12100890

**Published:** 2021-09-30

**Authors:** Huimin Guo, Benzheng Zhang, Xin Zheng, Juan Sun, Huiduo Guo, Gang Li, Guodong Zhao, Anying Xu, Heying Qian

**Affiliations:** 1College of Biotechnology, Jiangsu University of Science and Technology, Zhenjiang 212018, China; 15735531904@163.com (H.G.); zbzheng3399@163.com (B.Z.); a18796017350@163.com (X.Z.); sj15062884263@163.com (J.S.); guohuiduo1991@126.com (H.G.); gangsri@just.edu.cn (G.L.); sdgdzhao@126.com (G.Z.); xaysri@126.com (A.X.); 2The Sericultural Research Institute, Chinese Academy of Agricultural Sciences, Zhenjiang 212018, China

**Keywords:** *Bombyx mori* nuclear polyhedrosis virus, sequencing analysis, qPCR, pathogenicity difference

## Abstract

**Simple Summary:**

The hemolynamic septic disease in silkworms is caused by *Bombyx mori* nuclear polyhedrosis virus (BmNPV). It is the most severe viral disease that adversely affects the sericulture industry. Breeding BmNPV-resistant silkworm varieties is the most economic and effective solution. However, BmNPVs from different geographical strains have different pathogenicities. This brings the challenges of cultivating BmNPV-resistant silkworm varieties with wider adaptabilities. In this study, the genomes of two BmNPV strains (BmNPV ZJ and BmNPV YN) were sequenced and characterized to compare the difference in pathogenicity between the two strains. A total of 76 different genes in these two viruses were found with amino acid mutations. These included genes were associated with BmNPV replication and infection. In addition, the relative gene expression of the BmNPV YN strain was lower than that BmNPV ZJ. Thus, we speculate that the mutations in some genes may affect viral functions and may be the cause of the higher pathogenicity of BmNPV YN despite its lower proliferation rate. The present research provides new clues for further exploring the mechanism determining the difference in pathogenicity of different BmNPV strains.

**Abstract:**

The pathogenicity of different concentrations of *Bombyx mori* nuclear polyhedrosis virus- Zhenjiang strain (BmNPV ZJ) and Yunnan strain (BmNPV YN) was assessed in Baiyu larvae. The structures of the two viral strains were observed by negative-staining electron microscopy, and their proliferation was examined by quantitative polymerase chain reaction (qPCR). The genomic sequences of these two viruses were obtained to investigate the differences in their pathogenicity. The lethal concentration 50 (LC_50_) of BmNPV ZJ against Baiyu larvae was higher than that of BmNPV YN, indicating a relatively more robust pathogenicity in BmNPV YN. Electron microscopic images showed that the edges of BmNPV YN were clearer than those of BmNPV ZJ. The qPCR analysis demonstrated significantly higher relative expressions of immediately early 1 gene (*ie-1*), *p143*, *vp39*, and polyhedrin genes (*polh*) in BmNPV ZJ than in BmNPV YN at 12–96 h. The complete genomes of BmNPV ZJ and BmNPV YN were, respectively, 135,895 bp and 143,180 bp long, with 141 and 145 coding sequences and 40.93% and 39.71% GC content. Considering the BmNPV ZJ genome as a reference, 893 SNP loci and 132 InDel mutations were observed in the BmNPV YN genome, resulting in 106 differential gene sequences. Among these differential genes, 76 (including 22 hub genes and 35 non-hub genes) possessed amino acid mutations. Thirty genes may have been related to viral genome replication and transcription and five genes may have been associated with the viral oral infection. These results can help in understanding the mechanisms of pathogenicity of different strains of BmNPV in silkworms.

## 1. Introduction

*Bombyx mori* nuclear polyhedrosis virus (BmNPV), a circular double-stranded DNA virus, is the first insect baculovirus to be discovered [1,2]. The *B. mori* hemolynamic septic disease caused by BmNPV is the most severe viral disease in silkworms that harms the sericulture industry [3]. It is a subacute disease that occurs over 3–5 days, and the onset is faster in summer and autumn. The typical characteristics of infected silkworms include manic creeping, the body becoming whitish and shiny with intersegmental swelling, the skin easily breaking and oozing a milky white body fluid, and, finally, death. The milky white body fluid contains the BmNPV virus particles, which contaminate the mulberry leaves on which the silkworms reside. The contaminated leaf is eaten by other silkworms, which contract oral infections or infections through the wound after coming into contact with other wounded silkworms. The outbreak of the disease in rural production areas usually occurs in the middle and late stages of the fifth instar of *B. mori*. It is difficult to prevent and leads to a significant reduction in the quantity of production and, sometimes, no harvest occurs. Cultivating silkworm varieties with high resistance to BmNPV is the most economical and effective means to reduce the loss in sericulture production. Over the last 10 years, various institutions in China have cultivated several silkworm varieties resistant to BmNPV [4,5,6,7]; for example, “Huakang No.2”, which exhibits more than 100 times improved resistance to *B. mori* hemolynamic septic disease of the typical summer and autumn species “Qiufeng × Baiyu”. The silk productivity of these varieties also indicates their remarkable disease resistance properties [8,9,10,11].

Like in other viruses, genetic variation has also been found in BmNPV [12,13,14] and characterizes the different geographical strains [15,16]. In addition, different geographic strains of BmNPV have differences in their morphology and pathogenicity [17,18]. Even the virulence of local strains of BmNPV from the same province also differs against *B. mori*. Bai et al. [19] and Tang et al. [12] found that the pathogenicity of eight strains of BmNPV in samples from various regions in Yunnan differed and exhibited varying rates of infectivity in *B. mori*. Wang et al. [20] reported that the pathogenicity of BmNPV from different origins in Guangxi Province was also distinct in different silkworm varieties, which brought a new challenge to the breeding of silkworms against BmNPV. It is therefore necessary to explore the pathogenic mechanisms of different mutant strains of BmNPV in relation to silkworms.

We isolated two BmNPVs from Zhenjiang, Jiangsu Province, and Luliang, Yunnan Province, with varying pathogenicities to the same silkworm varieties. In the present study, we applied Illumina second-generation sequencing technology and Pacbio third-generation sequencing technology to characterize their genomes and explore the underlying mechanism of the variation in pathogenicity of different BmNPV strains.

## 2. Materials and Methods

### 2.1. Materials

BmNPV strains were obtained from different geographic regions: the BmNPV Zhenjiang strain (BmNPV ZJ) was obtained from Zhenjiang, Jiangsu province, and the BmNPV Yunnan strain (BmNPV YN) was obtained from Luliang, Yunnan province. These two strains were isolated, purified, and preserved by our teams.

The silkworm variety “Baiyu”, bred by the Institute of Sericulture of the Chinese Academy of Agricultural Sciences, was the parent of the control variety of silkworm varieties approved by the state for use in summer and autumn and was maintained in our laboratory.

### 2.2. Methods

#### 2.2.1. Virus Collection and Purification

The budded virus (BV) of BmNPV ZJ was preserved and provided by the Pathology Laboratory of the Institute of Sericulture of the Chinese Academy of Agricultural Sciences. The BV of BmNPV YN was preserved and provided by the Institute of Sericulture of the Yunnan Academy of Agricultural Sciences. In the autumn of 2019, our team used the BV to puncture the fifth instar of silkworms. After the onset of the disease, the blood of the infected silkworm was collected, filtered, and centrifuged three to four times to obtain the purified virus solution. Then, it was diluted with an appropriate amount of double-distilled water and counted using a hemocytometer. The concentrations of BmNPV ZJ and BmNPV YN were 2.45 × 10^9^ and 3.66 × 10^9^, respectively; they were stored at 4 °C for further use.

#### 2.2.2. Determination of Pathogenicity of Different BmNPV to *B. mori*

The original viral solutions of the two BmNPV strains were diluted with sterile water to a total of five concentrations, from 1 × 10^4^–1 × 10^8^, and smeared on mulberry leaves to feed the Baiyu larvae. The mulberry leaves with smooth surfaces were cut into small pieces (2.7 cm × 2.7 cm) using a punch, and 50 μL of different concentrations of polyhedral suspension was added dropwise on the upper surface of each leaf piece. After smearing evenly, these were fed to the second instar larvae of the silkworm, and leaves with the same volume of sterile water were used as blank controls. There were a total of 90 silkworm larvae per treatment, with three replicates (30 silkworms each), for each BmNPV concentration [20]. Four mulberry leaves carrying the virus were fed to each silkworm in each area at one time. After 24 h, the mulberry leaves carrying the virus were replaced with plain mulberry leaves with no virus. The growth and development of silkworms were observed every day, and the diseased silkworms were removed in time to avoid cross-infection. The number of dead silkworms that were infected with BmNPV was recorded after their third instar stage. The lethal concentration 50 (LC_50_) of Baiyu was calculated using the Statistical Product and Service Solutions (SPSS, https://www.ibm.com/cn-zh/analytics/spss-statistics-software accessed on 26 September 2021), and the pathogenicities of BmNPV YN and BmNPV ZJ were compared.

#### 2.2.3. Negative-Staining Electron Microscopy to Observe BmNPV Particles

The purified BmNPV ZJ and BmNPV YN were suspended on a copper mesh supported by a polyvinyl alcohol formaldehyde membrane (formvar membrane). The samples were stained with 2% phosphotungstic acid (pH 7.2) for 20 s. After washing and drying, the morphologies of the virus particles were observed with negative-staining electron microscopy.

#### 2.2.4. Sequencing of BmNPV ZJ and BmNPV YN Genomes

DNA was extracted from BmNPV YN and BmNPV ZJ according to the alkaline lysis method [12]. The concentrations of the extracted viral DNA were measured using the Qubit quantitative detector. The DNA quality was assessed using 1% agarose gel electrophoresis. The high-quality viral DNA was used for genome sequencing using Illumina second-generation and PacBio third-generation sequencing technologies (Sangon Biotech Co., Ltd., Shanghai, China).

#### 2.2.5. Analysis of BmNPV ZJ and BmNPV YN Genome Sequences

Using the genome of BmNPV T3 (L33180.1, 128,413 bp) as the reference, the sequencing data were assembled and corrected by SPAdes (https://cab.spbu.ru/software/spades/ accessed on 26 September 2021) and Primer-initiated Sequence Synthesis for Genomes (PrlnSeS-G, https://updeplasrv1.epfl.ch/prinses/ accessed on 26 September 2021) [21,22]. The Gene Ontology, Kyoto Encyclopedia of Genes and Genomes, Clusters of Orthologous Groups (COGs) of proteins, non-redundant proteins (NR), curated protein families (PFAM), Swiss-Prot, and TrEMBL databases were used for functional annotation of the virus genes. The genome sequences of these two BmNPVs were compared with other published BmNPV genome sequences. For multiple sequence alignment, the ClustalW parameters of Molecular Evolutionary Genetics Analysis (MEGA 7.1, https://megasoftware.net/ accessed on 26 September 2021) were used. The baculovirus repeated orfs gene (*bro)* of the BmNPV exists in diverse copies and nucleotide sequences in different ecological environments; therefore, this gene was also used for molecular identification of different strains of BmNPV [15,16,23]. The bootstrap statistical method was used for the calculation of 1000 replicates. The phylogenetic tree of BmNPV was constructed using the *bro-d* gene sequences of different BmNPV strains.

#### 2.2.6. Detection of Gene-Relative Expression of BmNPV by qPCR

The two strains of BmNPV were diluted to 1 × 10^8^ and fed to the fifth instar of the Baiyu silkworms, feeding 7 μL to each silkworm; there were five silkworms per treatment. The disease incidence rate among the silkworms was 100% at this concentration. At 12, 24, 48, 72, and 96 h, the silkworms were dissected and RNA was extracted from their midgut tissues. The very early gene *ie-1*, early gene *p143*, late gene *vp39*, and very late gene *polh* of BmNPV were chosen for quantitative polymerase chain reaction (qPCR) analysis. The primers for viral genes were designed in Primer 6.0 and their details are listed in Table 1. Reaction conditions for qPCR were: 95 °C, 30 s; 95 °C, 10 s; and 55 °C, 30 s, for a total of 40 cycles. Using *actin-3* (U49854) as the reference gene, the relative expression levels of the genes in the two BmNPV strains were calculated with the 2^−ΔΔCT^ method.

#### 2.2.7. BmNPV YN and BmNPV ZJ Differential Genes Analysis

Referring to the design process proposed by the Genome Analysis Toolkit (GATK) [24], the effective data for the BmNPV YN and BmNPV ZJ genomes were compared using the Burrows–Wheeler Aligner (BWA, http://bio-bwa.sourceforge.net/ accessed on 26 September 2021). To convert and sort the results, sequence alignment/map Tools (SAMtools, http://www.htslib.org/doc/samtools.html accessed on 26 September 2021) was used and the comparison results were statistically analyzed. The genotype differences between BmNPV YN and BmNPV ZJ were assessed using the HaplotypeCaller software, and the single nucleotide polymorphism (SNP) and insertion and deletion (InDel) information of the two strains were obtained. After quality control (Table 2), the SNP and InDel information were annotated using SNP effect software (SnpEff, https://pcingola.github.io/SnpEff/ accessed on 26 September 2021) [25]. According to the annotation results and BLAST analysis of the differential genes between BmNPV YN and BmNPV ZJ, candidate genes associated with the difference in pathogenicity of the BmNPV YN and BmNPV ZJ were screened.

## 3. Results

### 3.1. The Pathogenicity of BmNPV ZJ and BmNPV YN to B. mori

The mortality rate of Baiyu larvae increased with the increase in the concentration of BmNPV ZJ and BmNPV YN (Figure 1). At the same concentration, BmNPV YN exhibited stronger pathogenicity than BmNPV ZJ. In addition, the mortality rates at 1 × 10^7^ and 1 × 10^8^ of both BmNPVs differed significantly.

The median lethal dose of the two viruses in the Baiyu larvae was calculated by SPSS v21 (https://www.ibm.com/cn-zh/analytics/spss-statistics-software accessed on 26 September 2021) software. The LC_50_ concentration of BmNPV YN in Baiyu larvae was about 10 times less than that of BmNPV ZJ (Table 3), indicating that BmNPV YN could even cause death in half of these larvae at a low concentration.

### 3.2. The Morphology and Size of BmNPV ZJ and BmNPV YN

The purified BmNPV ZJ and BmNPV YN virus particle suspensions were observed by electron microscopy (Figure 2). The sizes of BmNPV ZJ and BmNPV YN appeared similar, with diameters of about 2.2–4.0 μm. The BmNPV polyhedra appeared mostly hexagonal, while a small number of particles appeared quadrilateral and irregular. The edges of BmNPV YN particles appeared sharper compared with those of BmNPV ZJ.

### 3.3. The Relative Expression of Genes of BmNPV ZJ and BmNPV YN in the Midgut of Baiyu

The qPCR analysis of the genes of BmNPV ZJ and BmNPV YN in the midgut of Baiyu larvae (Figure 3A) revealed that the relative expression of *ie-1* in BmNPV ZJ was higher than that in BmNPV YN at 12–96 h, with the highest level at 96 h. The expressions of *p143*, *vp39*, and *polh* in BmNPV ZJ were also higher than BmNPV YN before 96 h (Figure 3B–D).

### 3.4. Structural Characteristics of Virus Genomes

#### 3.4.1. Genome Characteristics of BmNPV ZJ

The genome of BmNPV ZJ was estimated to be 135,895 bp long, with 40.39% GC content. It comprised 141 predicted protein-coding sequences (CDSs), accounting for 83.86% of the entire genome sequence, of which 70 CDSs were on the positive strand and 71 on the negative strand. The average length of this genome was estimated to be 808 bp. Further analysis indicated 91 putative genes of 500 to 1000 bp lengths and 40 genes with ≥1000 bp lengths. Only 10 genes were less than 500 bp in length. In total, 139 genes were annotated. The annotated genes fibroblast growth factor (*fgf*), chi-tinase gene (*chi-a*), and alkaline exonuclease gene (*alk-exo*) were found in the BmNPV ZJ genome, but their homologous reading frames were absent in BmNPV YN. Further, late expression factor 12 (*lef-12*) and *AcMNPV 94K* were not annotated in the BmNPV T3 reference strain (Appendix A). Compared with the genome of BmNPV T3, some genes in these two viral genomes had SNP mutations, resulting in premature stop codons, and truncated or abnormal proteins. One putative gene with an unknown function was also found.

#### 3.4.2. Genome Characteristics of BmNPV YN

The genome of BmNPV YN was estimated to be 143,180 bp long, with 39.71% GC content. It comprised 146 CDSs, accounting for 89.37% of the whole genome sequence, of which 76 CDSs were on the positive strand and 69 on the negative strand. The average length of this genome was estimated to be 882.52 bp, with 96 genes of 500 to 1000 bp lengths and 43 genes more than 1000 bp in length. Six genes were found to be of less than 500 bp in length and 137 genes were annotated. Further analysis revealed *Orf20*, late expression factor 7 (*lef-7*), and *p26* genes in the BmNPV YN genome, but the homologous genes were absent in the BmNPV ZJ genome. *Lef-12* was not annotated in the BmNPV T3 reference strain (Appendix A). In addition, seven putative genes with unknown functions were found.

#### 3.4.3. Genome Comparison of BmNPV

The genome sequences of the two BmNPVs were compared with that of the BmNPV T3 strain as a reference, as shown in Table 4.

The nucleotide sequences of the BmNPV ZJ and BmNPV YN genomes were analyzed by DNAMAN software (https://www.lynnon.com/dnaman.html accessed on 26 September 2021). The genes of BmNPV ZJ and BmNPV YN were 92.6–100.0% homologous. A comparison with the genome sequences of the other six BmNPV reference strains showed homologies of BmNPV ZJ and BmNPV YN in the range of 92.0–100.0% and 93.3–100.0%, respectively (Appendix A). The *bro-d* gene sequence of BmNPV ZJ and BmNPV YN showed 95.9% homology, and the homologies with the other BmNPV reference strains were 92.1–99.6% and 93.3–97.3%, respectively (Table 5). The amino acid sequence homology of the BRO-D (encoded by the *bro-d*) of BmNPV ZJ and BmNPV YN was 96.6%, and the homologies with other BmNPVs were 92.0–99.7% and 92.2–97.1%, respectively (Table 5). The amino acid of the BRO-D protein of the two strains was mainly mutated at the N-terminal (Figure 4), which was consistent with the observation of Zhou et al. [26].

The *bro-d* gene was chosen for the evolutionary analysis of the two BmNPVs. Using *Plutella xylostella* multiple nucleopolyhedrovirus (PxMNPV) as the exogenous reference, the phylogenetic tree was plotted using the MEGA 7.1 software (https://megasoftware.net/ accessed on 26 September 2021). The results are shown in Figure 5. The eight BmNPV strains were divided into two clades. Both BmNPV ZJ (Zhenjiang) and BmNPV YN (Yunnan) were grouped in clade I, and BmNPV ZJ was closer to the Cubic strain (IQ991009) while BmNPV YN was closer to the India strain (JQ991010). These finding are similar to those reported by Tang et al. [12].

### 3.5. Detection and Analysis of Single Nucleotide Polymorphism and InDels in the Two Viral Genomes

Using BmNPV ZJ as the reference genome, a total of 893 SNP loci and 132 InDel variants were identified in BmNPV YN. The number of SNPs of transitions and transversions were 693 and 195, respectively. The SNP loci of the BmNPV YN strain were mainly enriched in the coding region, accounting for about 82.4% of the total number of SNPs. Among them, there were 487 synonymous mutations, 247 missense mutations, and 2 nonsense mutations, resulting in 102 different gene sequences (Table 6).

Analysis of InDel mutations by SnpEff software showed that the numbers of InDel mutations were similar between coding and non-coding regions (Table 7). Further analysis revealed 38 in-frame mutations and 25 frameshift mutations, resulting in 30 differential gene sequences.

### 3.6. Differential Genes Analysis

The BmNPV ZJ and BmNPV YN strains were observed to have 106 different annotated genes sequences, which included non-synonymous mutations, synonymous mutations, and frameshift mutations. Among these, 76 differential nucleotide sequences led to differences in amino acid sequences in the two viruses (Table 8). The core genes [27] (shared genes of the baculovirus) accounted for 28.94% and non-core genes accounted for 47.37% of all differential genes, while some genes had not been reported earlier. Thirty annotated genes were associated with the viral genome replication and transcription, and five genes were related to the oral infection of viruses. These differences in genes may have been the cause of the difference in pathogenicity between the two virus strains.

## 4. Discussion

BmNPV is a serious threat to silkworms and causes huge economic losses to the sericulture industry. Cultivating silkworm varieties resistant to a hemolymph-type septic diseases is the most effective and economical measure to minimize and mitigate these losses [28]. The pathogenicity of BmNPV varies in different geographic regions [18,19]. Furthermore, the underlying molecular mechanism that causes the variations in the pathogenicity of different strains of BmNPV remains unknown and poses a great challenge to the effective prevention of BmNPV. We compared the genomes of two BmNPV strains with different pathogenicities and found differences in some genes related to viral replication and infection.

In this study, the pathogenicities of BmNPV ZJ and BmNPV YN was determined and compared. The LC_50_ of BmNPV YN and BmNPV ZJ against Baiyu larvae were 3.62 × 10^5^ and 6.45 × 10^6^, respectively, indicating the semi-lethality of BmNPV YN in silkworms at a low concentration. Subsequently, the electron microscopic observation morphologies of virus particles revealed their hexagonal shapes with similar sizes (about 2.2–4.0 μm). These were typical silkworm nuclear polyhedrosis viruses, but the edges of BmNPV YN were clearer than those of BmNPV ZJ. The relative expressions of the *ie-1*, *p143*, *vp39*, and *polh* genes, expressed, respectively, at the very early, early, late, and very late stages of BmNPV ZJ, in the midgut of Baiyu larvae were higher than those of BmNPV YN within 12–96 h. This further indicates that the lethal concentration of BmNPV ZJ was higher than that of BmNPV YN. After a host is infected by a virus, in general, viruses with a high rate of proliferation tend to exhibit more robust pathogenicity. However, some more lethal virus strains may kill the host even with a lower rate of proliferation [29,30,31]. BmNPV YN might be such a type of lethal strain which, even in lower numbers, can kill the host. This inference is consistent with the finding of a tenfold lower LC_50_ for BmNPV YN than for BmNPV ZJ in the second instar of the Baiyu silkworm larvae. Therefore, when the virus was fed at a high concentration (10^8^) to silkworms, the expression of the virus gene of BmNPV YN was lower, but it was enough to kill the host. Likewise, Ma [32] found that the mutation and recombination in genomes of different strains of *Helicoverpa armigera* nucleopolyhedrovirus might be the cause of the difference in their pathogenicity even when they are distributed in the same geographic location. Hence, the difference in pathogenicity of our strains might have been because of the variation in their genomes due to the differences in the climates of Zhenjiang, Jiangsu Province, and Luliang, Yunnan Province, which are located, respectively, along the southeast coast and the inland southwest of China.

To further explore the reasons for the difference in pathogenicity between BmNPV YN and BmNPV ZJ to *B. mori*, the whole genomes of the virus strains were sequenced and analyzed. The genome size of BmNPV ZJ was estimated to be 135,895 bp, with 40.39% GC content and encoding 141 genes, while the genome size of BmNPV YN was estimated to be 143,180 bp, with 39.71% GC content and encoding 145 genes. These data demonstrated that the genome sizes of the two viruses and the numbers of putative genes changed. For further analysis, we used the *bro-d* gene, which is highly conserved in different strains of BmNPV from varying geographical regions. BmNPV ZJ exhibited the highest homology with the Cubic strains published earlier in the GenBank, which might be due to the closer origin region of BmNPV. These findings suggest that BmNPV from different regions must be used for screening the BmNPV-resistant silkworm varieties for their wider adaptability.

A comparative analysis of BmNPV YN and BmNPV ZJ genomic sequences revealed a large number of SNP and InDel mutations in the two genomes. The nucleotide sequence homology of the two virus genomes was 92.6–100.0%, and a further BLAST analysis revealed 76 different genes between BmNPV YN and BmNPV ZJ. These genes included 22 core genes and 36 non-core genes, mainly the oral infection factor genes, viral replication genes, host molting, and pupation genes. The alterations of genes sequences affect the function of the encoded proteins. Therefore, we speculated that the difference in the genomes of the two viruses might have been the cause of the difference in their pathogenicity in silkworms. Thus, the roles of viral genes and the mechanism of action of important proteins encoded by these genes need to be studied in the future.

## 5. Conclusions

To summarize, the BmNPV YN strain has a relatively lower rate of proliferation but stronger pathogenicity than BmNPV ZJ. The two viruses differ in terms of genome size and the number of coding genes. Among these, 76 genes are different, including genes associated with BmNPV virus replication and infection. These data illustrate that, due to different geographical environments, the genomes of different BmNPV strains mutate and rearrange, which finally leads to differential pathogenicity. The results of this study can provide references to further explore the molecular mechanism relating to the difference in pathogenicity among the viral strains and also provide some guidance in developing new insecticides.

## Figures and Tables

**Figure 1 insects-12-00890-f001:**
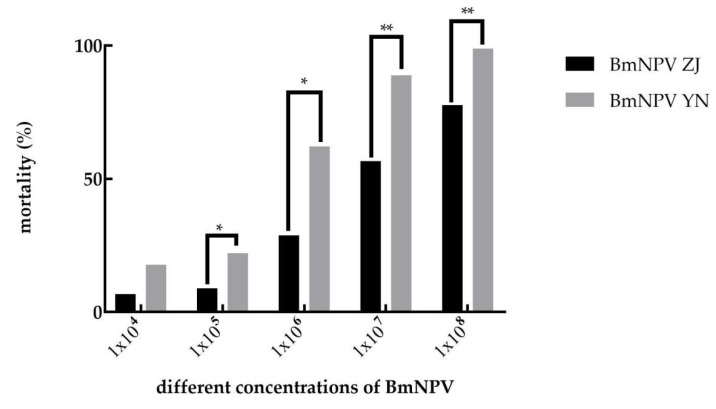
A comparison of the pathogenicities of different concentrations of the two BmNPV strains against Baiyu larvae. Note: “*” represents *p* < 0.05; “**” represents *p* < 0.01.

**Figure 2 insects-12-00890-f002:**
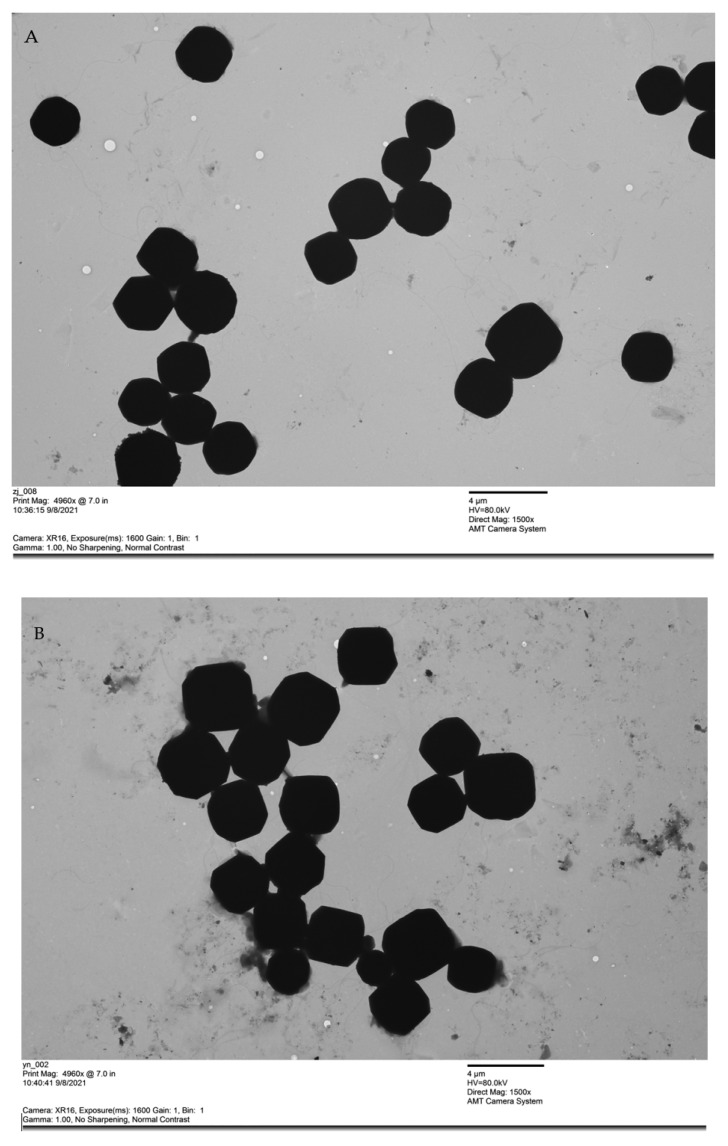
Morphological observation of BmNPV polyhedra by electron microscopy (×1500): (**A**) BmNPV ZJ and (**B**) BmNPV YN.

**Figure 3 insects-12-00890-f003:**
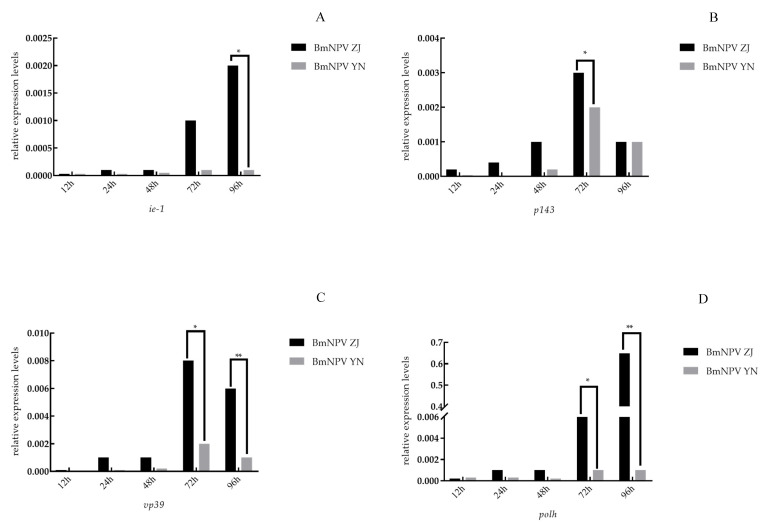
The relative expression of genes of the two BmNPV strains in the midgut of Baiyu larvae: (**A**) *ie-1*; (**B**) *p143*; (**C**) *vp39*; (**D**) *polh.* Note: “*” represents *p* < 0.05; “**” represents *p* < 0.01.

**Figure 4 insects-12-00890-f004:**
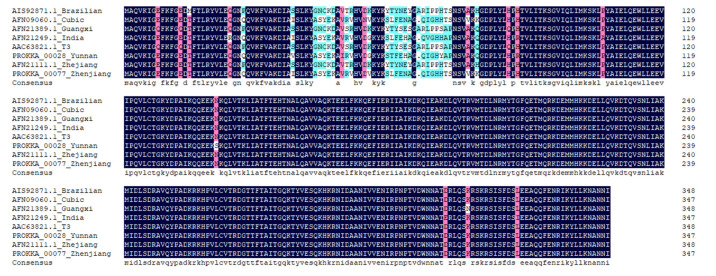
Alignment of amino acid sequences of BRO-D protein from different BmNPV strains.

**Figure 5 insects-12-00890-f005:**
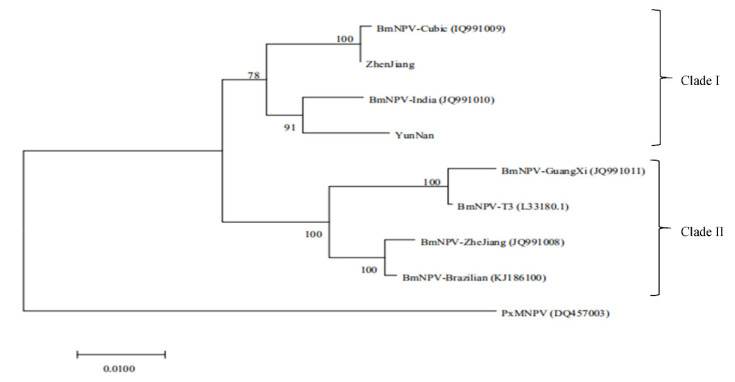
Phylogenetic tree constructed based on bro-d gene sequences of BmNPV isolates.

**Table 1 insects-12-00890-t001:** Primers used in qPCR.

Gene	Primer Sequences	Length/(bp)
*actin-3*	F: 5′-CGGCTACTCGTTCACTACC-3′	147
	R: 5′-CCGTCGGGAAGTTCGTAAG-3′	
*ie-1*	F: 5′-GGACGAATACTTGGACGAT-3′	237
	R: 5′-GAGAACCTGTTGGAATTGTAG-3′	
*p143*	F: 5′-GCACGGCAATACTTATCATC-3′	120
	R: 5′-TGAGCACCAACAATAGTCC-3′	
*vp39*	F: 5′-ACACGGAGGAATTGAGATT-3′	116
	R: 5′-GATGTCACTGCTTCTTATCG-3′	
*polh*	F: 5′-CTACAAGTTCCTCGCTCAA-3′	163
	R: 5′-CTCGCTGTGGATGTTCAT-3′	

**Table 2 insects-12-00890-t002:** SNP and InDel quality control standards.

Quality Control Items	SNP	InDel
QualByDepth	≥2.0	≥2.0
RMSMappingQuality	≥40.0	-
FisherStrand	≤60.0	≤200.0
StrandOddsRatio	≤3.0	≤10.0
MappingQualityRankSumTest	≥−12.5	≥−12.5
ReadPosRankSumTest	≥−8.0	≥−8.0

**Table 3 insects-12-00890-t003:** Comparison of virulence of BmNPV ZJ and BmNPV YN in Baiyu larvae.

Variety	Strain	Regression Equation	LC_50_	95% Confidence Value	Slope of Regression Line/SE
Baiyu	BmNPV ZJ	y = −4.267 + 0.627x	6.45 × 10^6^	3.88 × 10^6^–1.14 × 10^7^	0.056
Baiyu	BmNPV YN	y = −4.426 + 0.796x	3.62 × 10^5^	1.98 × 10^5^–6.45 × 10^5^	0.063

**Table 4 insects-12-00890-t004:** General features of BmNPV genomes.

Items	BmNPV ZJ	BmNPV YN	BmNPV T3
Size (base)	135,895	143,180	128,413
G + C content (%)	40.39	39.71	40
Protein coding genes	141	145	136
Min length (base)	111	111	61
Max length (base)	2964	4050	1222
Average length (base)	808.23	882.52	852.154
Total coding gene (base)	113,961	127,965	115,893
Coding ratio (%)	83.86	89.37	90.25

**Table 5 insects-12-00890-t005:** Nucleotide and amino acid homology of *bro-d* gene sequences.

Reference Strain	Nucleotide Homology/(%)		Reference Strain	Amino Acid Sequence/(%)	
	BmNPV ZJ (N)	BmNPV YN (N)		BmNPV ZJ (AA)	BmNPV YN (AA)
Brazilian	94.3	94.2	Brazilian	93.7	92.5
Cubic	99.6	95.7	Cubic	99.7	96.3
Guangxi	92.1	93.3	Guangxi	91.7	92.2
India	96.8	97.3	India	96.6	97.1
T3	95.9	94.1	T3	92.0	92.5
Zhejiang	95.5	94.3	Zhejiang	94.0	92.8
Zhenjiang	-	95.9	Zhenjiang	-	95.9
Yunnan	96.9	-	Yunnan	96.6	-

**Table 6 insects-12-00890-t006:** Summary statistics of SNP base changes of BmNPV YN.

Type	SNP Number	Region
Transitions	693	
Transversions	195	
Mutations in coding region	736	
Intergenic mutation	151	
Synonymous mutation	487	CDS
Missense mutation	247	CDS
Nonsense mutation	2	CDS
Other mutations that could not be accurately judged	6	

Note: The reference genome was BmNPV ZJ.

**Table 7 insects-12-00890-t007:** InDel annotations of BmNPV YN.

Type		InDel Number	Region
Mutations in coding region		66	
Intergenic mutation		64	
Codon mutation	Code insertion	19	CDS
	Code deletion	19	CDS
Frameshift mutation		25	CDS
Other mutations that could not be accurately judged		6	

Note: The reference genome was BmNPV ZJ.

**Table 8 insects-12-00890-t008:** Viral functional genes with non-synonymous mutations and frameshift mutations.

Type	BmNPV ZJ Gene Number	BmNPV YN Gene Number	Gene Name	Mutation Type	Biological Function	Description
Core	PROKKA 00060	PROKKA 00011	*pif-2*	Transversions	Viral infection	The composition of the membrane, necessary for oral infection
	PROKKA 00082	PROKKA 00117	*p74* (*pif-0*)	Transversions	Viral infection	It is related to the life cycle of the virus and participation in the oral infection of the virus
	PROKKA 00108	PROKKA 00135	*pif-1*	Transitions and transversions	Viral infection	The composition of the membrane, related to oral infections
	PROKKA 00113	PROKKA 00108	*pif-5* (*odv-e56*)	InDel	Viral infection	Determines the virus host range, related to oral infections
	PROKKA 00009	PROKKA 00056	*Vlf-1*	Transversions and InDel	Replication, transcription	Late gene expression
	PROKKA 00020	PROKKA 00067	*dna pol*	Transitions and transversions	Replication, transcription	The catalytic activity, replication of the viral genome, and host DNA polymerase cannot replace viral enzymes in this process
	PROKKA 00035	PROKKA 00082	*lef-8*	Transitions	Replication, transcription	Late gene expression
	PROKKA 00043	PROKKA 00090	*p47*	Transitions	Replication, transcription	Regulation of viral transcription
	PROKKA 00067	PROKKA 00018	*lef-1*	InDel and transversions	Replication, transcription	Encoding DNA promoter and interacting with LEF-2
	PROKKA 00073	PROKKA 00024	*lef-2*	Transitions	Replication, transcription	Virus replication and late gene expression
	PROKKA 00092	PROKKA 00036	*lef-5*	Transitions	Replication, transcription	Regulation of viral transcription
	PROKKA 00095	PROKKA 00033	*AcMNPV orf103*	Transitions	Replication, transcription	Virus replication, influences virus particle formation
	PROKKA 00114	PROKKA 00109	*ie-1*	Transitions	Replication, transcription	The essential transactivated protein that initiates viral DNA replication and *bro* promoter transcription
	PROKKA 00123	PROKKA 00045	*lef-4*	Transitions and transversions	Replication, transcription	Regulation of viral transcription
	PROKKA 00128	PROKKA 00040	*dna hel/p143*	Transversions	Replication, transcription	DNA helicase activity, host domain determinant
	PROKKA 00132	PROKKA 00126	*p24*	Transitions	Replication, transcription	Regulation of viral transcription
	PROKKA 00003	PROKKA 00050	*p95*	Transitions	Structural protein	Composition of BV and ODV
	PROKKA 00006	PROKKA 00053	*gp41*	Transitions and transversions	Structural protein	Exists only in ODV, determining the manner and the ability of the virus to invade the host
	PROKKA 00008	PROKKA 00055	*AcMNPV orf78*	Transitions	Structural protein	Related to BV production and M-ODV formation
	PROKKA 00093	PROKKA 00035	*p40*	Transitions, transversions, and InDel	Structural protein	Includes body virus envelope components related to specific infection of host cells
	PROKKA 00100	PROKKA 00143	*AcMNPV orf109*	Transversions	Structural protein	Participates in viral nucleocapsid assembly
	PROKKA 00122	PROKKA 00046	*vp39*	Transitions and InDel	Structural protein	Related to virus infection
Non -core	PROKKA 00101	PROKKA 00142	*AcMNPV orf110*	Transitions	Viral infection	Related to oral infections
	PROKKA 00024	PROKKA 00071	*fp25K* (*ac61*)	InDel	Auxiliary function	Involved in BV and ODV formation, implicated in host degradation after death
	PROKKA 00058	PROKKA 00009	*pkip*	Transitions	Auxiliary function	Related to the BV nucleocapsid component
	PROKKA 00066	PROKKA 00017	*ecdysteroid UDP-glucosyl transferase* (*egt*)	Transversions	Auxiliary function	Hinders larvae molting and pupation
	PROKKA 00070	PROKKA 00021	*pk1*	Transversions and InDel	Auxiliary function	Regulation of polyhedrin gene promoter activity
	PROKKA 00078	PROKKA 00029	*ptp*	Transitions	Auxiliary function	BV components, essential components for effective infection of larvae brain tissue
	PROKKA 00086	PROKKA 00129	*chitinase A*	Transitions	Auxiliary function	Related to virus transmission
	PROKKA 00140	PROKKA 00128	*viral cathepsin-like protein* (*v-cath*)	Transversions	Auxiliary function	Related to host liquefaction and degradation
	PROKKA 00016	PROKKA 00063	*AcMNPV orf69*	Transitions	Replication, transcription	Late gene expression
	PROKKA 00018	PROKKA 00065	*lef-3*	Transitions	Replication, transcription	SS-DNA binding and destruction of helical stability
	PROKKA 00031	PROKKA 00078	*lef-10*	Transitions	Replication, transcription	Regulation of viral transcription
	PROKKA 00042	PROKKA 00089	*lef-12*	Transitions	Replication, transcription	Late gene transcirition
	PROKKA 00052	PROKKA 00002	*bro-a*	Transitions and transversions	Replication, transcription	DNA binding protein, complementary to BRO-C
	PROKKA 00054	PROKKA 00005	*lef-6*	Transitions and transversions	Replication, transcription	Virus replication and late gene expression, affecting host cell apoptosis
	PROKKA 00057	PROKKA 00008	*dbp* (*DNA binding protein*)	InDel	Replication, transcription	SS-DNA binding protein co-localized with *ie-1* and *lef-3* in viral replication mechanism
	PROKKA 00069	PROKKA 00020	*bm* (*br*) *orf-4*	Transitions and transversions	Replication, transcription	Early gene expression of virus
	PROKKA 00071	PROKKA 00022	*orf1629*	Transitions and transversions	Replication, transcription	Virus replication
	PROKKA 00076	PROKKA 00027	*Bro-b*	Transitions, transversions, and InDel	Replication, transcription	DNA binding protein
	PROKKA 00077	PROKKA 00028	*Bro-d*	Transitions, transversions, and InDel	Replication, transcription	Virus replication and gene expression regulation
	PROKKA 00080	PROKKA 00105	*pe38*	Transitions and transversions	Replication, transcription	Virus replication and gene expression regulation
	PROKKA 00090	PROKKA 00038	*Bro-c*	Transitions, transversions, and InDel	Replication, transcription	DNA binding protein, complementary to BRO-A
	PROKKA 00091	PROKKA 00037	*39k*	InDel	Replication, transcription	Virus replication and gene expression regulation
	PROKKA 00097	PROKKA 00030	*he65*	Transitions and transversions	Replication, transcription	Virus replication
	PROKKA 00111	PROKKA 00106	*ie-2*	InDel	Replication, transcription	Virus replication and gene expression regulation
	PROKKA 00120	PROKKA 00115	*ie-0*	Transitions	Replication, transcription	Regulation of viral transcription
	PROKKA 00121	PROKKA 00116	*me53*	Transitions and transversions	Replication, transcription	Related to BV and ODV production
	PROKKA 00021	PROKKA 00068	*gp37*	Transitions	Structural protein	The formation of auxiliary components of polyhedra is involved in the transport of virus particles in the cell
	PROKKA 00037	PROKKA 00084	*odv-e66*	Transitions and transversions	Structural protein	Participation in BV and ODV morphogenesis and oral infection
	PROKKA 00038	PROKKA 00085	*bmnpvcubigcp037*	InDel	Structural protein	Replication of the virus, regulation gof the transport of virus particles
	PROKKA 00065	PROKKA 00016	*Bv/odv-e26*	Transitions	Structural protein	Related to BV and ODV envelopes
	PROKKA 00096	PROKKA 00032	*vp80*	Transitions	Structural protein	Required for virus replication, BV production, and nucleocapsid maturation
	PROKKA 00135	PROKKA 00123	*AcMNPV orf132*	Transitions and transversions	Structural protein	Involved in BV and ODV formation
	PROKKA 00139	PROKKA 00118	*p10*	Transitions	Structural protein	Partcipates in the morphogenesis of viral polyhedra and promotes the release of polyhedra from infected insect cells
	PROKKA 00015	PROKKA 00062	*iap2*	Transitions	Apoptosis	Cell apoptosis inhibiting factor
	PROKKA 00055	PROKKA 00006	*iap1*	Transitions, transversions, and InDel	apoptosis	Induction of apoptosis

## Data Availability

No new data were created or analyzed in this study. Data sharing is not applicable to this article.

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
