# Peer review of "Pathogenicity Detection and Genome Analysis of Two Different Geographic Strains of BmNPV"

_insects, 2021, doi:10.3390/insects12100890_

Round 1

Reviewer 1 Report

This study seeks to understand genetic mechanisms that underlie variance in Bombyx mori nuclear polyhedrosis virus (BmNPV) pathogenicity to the silkworm Bombyx mori to aid breeding programs of high resistance B. mori strains. It seems that all methods and analyses have been carefully and correctly carried out, but some details are missing. The results show differences in both genomic structure and expression patterns between these two virus strains directing future research to explore the mechanisms of pathogenicity.

Comments:

The introduction could be more thorough, e.g. describing what kinds of symptoms the insects get from the virus, how much mortality the virus causes, is the problem seasonal, regional etc. In addition, it would be nice to include a more precise description on the resistant silkworm varieties.

Please provide detailed information about the isolation, purification and preservation methods of the two BmNPV strains. In addition, describe more clearly what is meant by treatment and more details on how the different silkworm numbers per treatment are derived.

Please explain why bro-d gene was selected for phylogeny reconstruction of BmNPV.

Were biological and technical replicates included in the qPCR? How many?

Please provide more details on how SNPs and indels were determined (e.g. were filtering for read and mapping quality used prior to this?). I suggest that procedures with GATK and SnpEff would be described separately and with more detail. Similarly, I suggest that analysis with BWA and Blast would be described separately and in more detail.

Figure 1 legend contains errors – please correct.

Was any mortality observed in the controls?

Figure 2 is not readable.

Figure 3 and 4 need more explanation to be understandable.

BmNPV YN contained 5 ORFs more than BmNPV ZJ: does this mean that YN contained all the genes present in ZJ and 5 more, or how did the gene contents overlap? Were the five functionally unknown genes the same in these two strains?

It would have been nice if a pairwise alignment of these two virus genomes would have been performed and structural differences visualized in a figure.

Line 322: please describe ‘core gene’ and ‘non-core gene’.

Please indicate in table 5 which genes contained each type of mutations.

The discussion could be more structured and organized, now it is a bit messy and difficult to follow. A bit more discussion on the quite drastic gene expression differences between these two strains would be nice.

Quite many citations have returned errors (reference source not found).

The whole text needs language and grammar revision as it is not fully understandable now. In addition, I have listed below some passages of text that are not clear and need revision:

Line 20: encodes by partial differential genes of the two viruses

Line 36: 76 genes involved in amino acid

Line 52: has the same genetic variation as other virus

Line 74: Baiyu were provided by our laboratory (what is Baiyu?)

Line 85: what is meant by the district?

Line 110-111: Please clarify these sentences.

Line 347-350: gene expression is higher in ZJ, but does it directly mean that the lethal concentration of ZJ is higher than YN? Rather it seems that ZJ is more active at these time points than YN.

Author Response

Dear Reviewer:

Thank you for your comments concerning our manuscript entitled “Pathogenicity detection and genome analysis of two different geographic strains of BmNPV” (insects-1337816). Those comments are all valuable and very helpful for revising and improving our paper, as well as the important guiding significance to our researches. We have studied comments carefully and have made correction which we hope meet with approval. Revised portion are marked in yellow in the paper. The main corrections in the paper and the responds to the reviewer’s comments are as flowing:

  1. Response to comment: The introduction could be more thorough.

Response: According to your comments, we have made corresponding supplements to the introduction (Line 48-58, page 2).

  1. Response to comment: Please provide detailed information about the isolation, purification and preservation methods of the two BmNPV strains.

Response: According to your comments, we have added “virus collection and purification” in the “Methods” section (Line 92-101, page 2-3).

  1. Response to comment: Describe more clearly what is meant by treatment and more details on how the different silkworm numbers per treatment are derived.

Response: I am sorry that this part was not clear in the original manuscript. I should have explained that the number of silkworms selected during our experiment was selected after referencing some articles.. I have revised the contents of this part (Line 109-111, page 3).

The reference was“Wang, X.; Huang, X.H.; Jiang, M.G.; et al. Epidemic factors of Bombyx mori hemolymph-type septic disease in Guangxi and its molecular phylogenetic analysis. J South Agric 2020, 051, 669-676.”

  1. Response to comment:Explain why bro-dgene was selected for phylogeny reconstruction of BmNPV.

Response: We selected bro gene because it was clear that bro gene could be used as the basis for BmNPV classification and identification of different strains after previous attempts by scholars. Therefore, we selected bro gene to draw the phylogenetic tree. I have made corresponding supplement in “2.2.4. Sequence analysis of different BmNPV” (Line 131-136, page 3).

Tang, F.F.; Shao, X.L.; Zhong, J.; et al. A preliminary study on molecular identification of Bombyx mori nucleopolyhedrovirus strains. Sci Seric 2014, 1030-1035.

Wonkyung K, Masataka S, Evgueni Z, et al. Characterization of Baculovirus Repeated Open Reading Frames (bro) in Bombyx moriNucleopolyhedrovirus. J. Virol., 1999. 73(12): p. 10339-10345.

  1. Response to comment: The replicates of biological and technical included in the qPCR.

Response: We did five biological replicates and three technical replicates in the qPCR, and we added the corresponding description in the article (Line 141, page 3).

  1. Response to comment: Provide more details on how SNPs and indels were determined.

Response: I am sorry that this part was not clear in the original manuscript. Related experiment process and SNP, InDel quality control standards have been added to “2.2.6. Differential genes analysis between BmNPV YN and BmNPV ZJ” (Line 149-156, page 4).

  1. Response to comment: Figure 1 legend contains errors – please correct.

Response: We are sorry that the picture and the corresponding description have shifted due to our negligence. We have changed the part of Figure 1 (Line 178-179, page 5).

  1. Response to comment: Was any mortality observed in the controls?

Response: We have smeared the same amount of mulberry leaves with double distilled water as a control. In the control group, no silkworm died due to BmNPV.

  1. Response to comment: Figure 2 is not readable.

Response: We have re-edited to the Figure 2.

  1. Response to comment: Explain the Figure 3 and Figure 4

Response: Considering that there is little effective information in Figure 3 and figure 4, we decide to delete Figure 3 and Figure 4. Accordingly, we mapped the genome linear map of BmNPV ZJ and BmNPV YN to show the genome information of the two viruses. The pictures in the article will be displayed as supplementary materials (Figure S1, Figure S2).

  1. Response to comment: BmNPV YN contained 5 ORFs more than BmNPV ZJ: does this mean that YN contained all the genes present in ZJ and 5 more, or how did the gene contents overlap?

Response: The overlap of genes between the two strains is also annotated in our genome map (Figure S1, Figure S2).

  1. Response to comment: Were the five functionally unknown genes the same in these two strains?

Response: The genes with unknown functions of the two viruses are not the same. In fact, during our comparison using NBCI, we found that genes with unknown function of BmNPV ZJ have higher homology with AcMNPV; however, genes with unknown function of BmNPV YN have extremely low homology with the virus.

  1. Response to comment: Performed the structural differences between the two virus.

Response: Thank you for your comments, this could enrich our article. In fact, we are doing follow verification tests on the differential genes of the two viruses, and we are more inclined to show the specific structural differences between genes in the specific description of the function of a certain differential gene.

  1. Response to comment: Line 322: please describe ‘core gene’ and ‘non-core gene’.

Response: We have supplemented the corresponding definitions and references (Line 299-300, page 9).

  1. Response to comment: Please indicate in Table 5 which genes contained each type of mutations.

Response: We replace the evalue in Table 8 with the mutation type.

  1. Response to comment: Modification of the discussion section.

Response: We have re-written this part according to your suggestion (Line 335-341, page 14), (Line 347-357, page 14).

  1. Response to comment: Quite many citations have returned errors.

Response: I am very sorry for our negligence, and we have corrected the errors cited parts in the article (Line 143,page 3), (Line 162,page 4) , (Line 181, page 5), (Line 186 and 190, page 5), (Line 221, page 7), (Line 230, 232 and 234, page 7), (Line 254, page 8), (Line 287 and 299, page 9).

  1. Response to comment: The whole text needs language and grammar revision.

Response: According to your suggestions, we send the article to the company for revision and polishing. Unfortunately, the time limit for our article modification is only 10 days, so there is no way to submit a well-polished article. We can only submit the modified and unpolished version first. After the article has been polished, we will submit the polished version as soon as possible.

  1. Response to comment: Explain “Line 20: encodes by partial differential genes of the two viruses” and “ Line 36: 76 genes involved in amino acid”.

Response: I am sorry that this part was not clear. I should have explained that this sentence is based on the results of our SNP annotation, we found that the mutation of some genes' nucleotides caused their amino acid changes. We have also made corresponding changes in line 20 and line 36 of page 1 of the article

  1. Response to comment: Explain “Line 52: has the same genetic variation as other virus”.

Response: Some scholars have found that the genetic material of other viruses will change in the direction conducive to their survival during reproduction, which is consistent with our sequencing results. The most typical example is the COVID-19. In a relatively short period of time, genetic material mutate, which eventually leads to a gradual increase in the pathogenicity of the virus.

  1. Response to comment: What is Baiyu?

Response: Baiyu is a Japanese parent cultivated for summer and autumn by the Institute of Sericulture, Chinese Academy of Agricultural Sciences, and is preserved in our laboratory. We  also have added this imformation in the “Materials” section (Line 88-90, page 2).

  1. Response to comment: What is meant by the district?

Response: I am sorry that this part was not clear in the original manuscript. I should have explained that the district is silkworm area. In this experiment, the Baiyu we used was single cross hybridd. Therefore, the area in the article refers to a small group produced by the mating of the same male and female moths. I have revised the contents of this part (Line 111, page 3).

  1. Response to comment: Line 110-111: Please clarify these sentences.

Response: We have modified this part (Line 138-140, page 3)

  1. Response to comment: Line 347-350: gene expression is higher in ZJ, but does it directly mean that the lethal concentration of ZJ is higher than YN?

Response: The gene we use for qPCR is a single copy of the virus, and high gene expression means high virus concentration; and the preliminary experiments show that the virus concentration we feed is the lethal concentration of the silkworm.

We tried our best to improve the manuscript and made some changes in the manuscript. These changes will not influence the content and framework of the paper. Revised portion are marked in yellow in the paper.

We appreciate for your warm work earnestly, and hope that the correction will meet with approval.

Once again, thank you very much for your comments and suggestions.

Reviewer 2 Report

The study by Gun et al. sequenced and analyzed two BmNPV isolates. The overall design and flow seems fine but a significant improvement should be done before it can be accepted for publication.

1) The introduction section is too short and not such informative fro readers to understand the logic and importance of this study, especially the complexity of the baculovirus infection in insects.

2) One concern is that how the authors confirm the isolates are BmNPV but not AcMNPV. Is there some SEM/TEM verification?

3) BmNPV T3: not 141 gene? see Ono et al., 2012,  10.1016/j.virusres.2012.02.016. The other information is also not accurate for Table 2. Acreage length for T3 can be also calculated.

4) Fig. 3 and Fig. 4 are not helpful at all. Alternatively, the author should reconstruct the list (SNPs/Indel) to show the different genes/proteins in those strains using T3 as reference. Are the "mutated" sites essential for protein function?

5) Together with the Q4, Fig. 5 only shows the alignment of bro-d gene. The authors need also provide alignment results for other coding genes with non-synonymous SNPs/Indel as supplemental information. Among those, the authors could then pick up some essential genes, such as bro-d, lef-1, ie-1.... as representative results in main text. Specifically, the authors should focus on the factors related to oral infection and virus particle/OB production.

6) The resolution for figs should be improved in final version.

7) For fig. 2, how about the OB number?

8) What are the conclusions and significance of this study based on the results of this study? 

Author Response

Dear Reviewer:

Thank you for your comments concerning our manuscript entitled “Pathogenicity detection and genome analysis of two different geographic strains of BmNPV” (insects-1337816). Those comments are all valuable and very helpful for revising and improving our paper, as well as the important guiding significance to our researches. We have studied comments carefully and have made correction which we hope meet with approval. Revised portion are marked in yellow in the paper. The main corrections in the paper and the responds to your comments are as flowing:

  1. Response to comment: The introduction could be more thorough.

Response: According to your comments, we have made corresponding supplements to the introduction (Line 48-58, page 2) (Line 61-64, page 2).

  1. Response to comment: How the authors confirm the isolates are BmNPV but not AcMNPV?

Response: Thanks for reminding. In fact, these viruses are virus strains preserved by the pathology laboratory of the professional research institute, and the professionalism of the researchers ensures the purity of the virus. In addition, we conducted a pre-infection experiment, and the typical BmNPV infection characteristics appeared in the pathogenesis and symptoms of B. mori.

  1. Response to comment: BmNPV T3: not 141 gene?.

Response: Thanks for reminding. We have searched your suggested references, which has played a great help to us. However, considering that some of the genes used in this article are transcriptome sequencing results, we finally decided to refer to the genome information of BmNPV T3 in NCBI ( L33180.1 ).

  1. Response to comment: The other information is also not accurate in Table 2.

Response: I am sorry that this part was not clear in the original manuscript. We have changed the part of Table 4.

  1. Response to comment: Fig. 3 and Fig. 4 are not helpful at all.

Response: Considering that there is little effective information in Figure 3 and figure 4, we decide to delete Figure 3 and Figure 4. Accordingly, we mapped the genome linear map of BmNPV ZJ and BmNPV YN to show the genome information of the two viruses. The pictures in the article will be displayed as supplementary materials (Figure S1, Figure S2).

  1. Response to comment: Reconstruct the list (SNPs/Indel) to show the different genes/proteins in those strains using T3 as reference.

Response: Thank you for your suggestions. In the Figure S1 and Figure S2, we show which genes of the two viruses are new genes using T3 as a reference. However, considering that the purpose of the article is to explore the genomic differences between BmNPV ZJ and BmNPV YN, the SNP and InDel mutations compared to BmNPV T3 are not shown in the article.

  1. Response to comment: Are the "mutated" sites essential for protein function?

Response: Thank you for your valuable questions and ideas. In fact, the question you asked is exactly the experiment we are currently conducting to verify. However, the number of different genes between the two viruses exceeded our expectations. So I’m very sorry that we still have no way to give you a clear answer. But we have added relevant research in the article (Line 335-337, page 14).

  1. Response to comment: Provide alignment results for other coding genes.

Response: As you suggested, the alignment of bro-d may be fortuitous. Therefore, we compared the nucleotide sequences of the two virus strains with other BmNPV reference strains and revised the article (Line 225-227, page7). The comparison results are shown in the Table A1 in the article.

  1. Response to comment: The resolution for figs should be improved in final version.

Response: I am sorry that this part was not clear in the original manuscript. We replaced the unclear pictures, especially Figure 2.

  1. Response to comment: How about the OB number?

Response: Thank you for your valuable questions and ideas. Due to the lack of experimental equipment, we mailed our samples to Nanjing Agricultural University as soon as possible. Unfortunately, starting from July 20th, due to the sudden spread of the COVID-19, Nanjing city is still in a closed state so far, we are temporarily unable to conduct electron microscope observations. If necessary, we will update the manuscript as soon as the results are obtained.

  1. Response to comment: The conclusions and significance of this study based on the results of this study?

Response: I am sorry that this part was not clear in the original manuscript. The conclusions of this study is the differences in geographical environment, the genomes of different BmNPV strains were mutated and rearranged, and finally showed the difference in viral pathogenicity (Line 368-371, page 15). The significance of this study is that it can provide some clues and references for futher exploring the molecular mechanism that causes the difference in viral pathogenicity among different strains, and also provide some help for the development of new insecticides (Line 371-374, page 15).

We tried our best to improve the manuscript and made some changes in the manuscript. These changes will not influence the content and framework of the paper. Revised portion are marked in yellow in the paper.

We appreciate for your warm work earnestly, and hope that the correction will meet with approval.

Once again, thank you very much for your comments and suggestions.

Reviewer 3 Report

In the present study, Guo et al sequence and assemble two BmNPV genomes of differing virulence to compare the genomic content. The study is relatively simple and clearly lays out the differences between the genomes, however I feel it lacks a lot of content & does not perform a lot of the comparisons that could be made. I feel the manuscript could be improved if the authors went more in depth into the differences in gene content between the genomes. What are the genes which differ? what appears to be the source of these genes? Are they orthologous to anything? Could they play a role in differences in virulence (based on loci, if they landed upstream of an important gene). Currently the authors do not clearly answer their presented question & do not attempt to answer it from different perspectives. I would like to see this in the results & discussion in a resubmitted version of this manuscript.

Along with this I have several minor issues:

Line 114: Citation here is missing. This error is seen throughout the manuscript & should all be fixed before resubmitting.

Figure 1: Is there any comparison that can eb made to a reference strain from elsewhere? For people not knowledgeable in BmNPV to understand differences in virulence. Also, error in description: concentraTable 21

Figure 2: Error in labels of barplots, the A, B, C, D is covering one of the legends.

Figure 3 & 4: In the annotated genes, it would be appreciated if you could label which genes are orphans, which are shared with the other BmNPV & which are conserved Baculovirus genes. A supplementary figure or table showing the annotation of genomes would be incredibly helpful.

Figure 3 & 4: A ribbon plot connecting contiguous regions of these two genomes & to a BmNPV reference genome would be incredibly helpful for understanding how these genomes differ.

Line 223: empty parentheses?

Figure 5: Clearer labelling would be appreciated. Maybe a matrix of how similar each genome is would be more useful as a subpart of this figure, instead of table 4.

Line 313: SNPEff citation.

Author Response

Dear Reviewer:

Thank you for your comments concerning our manuscript entitled “Pathogenicity detection and genome analysis of two different geographic strains of BmNPV” (insects-1337816). Those comments are all valuable and very helpful for revising and improving our paper, as well as the important guiding significance to our researches. We have studied comments carefully and have made correction which we hope meet with approval. Revised portion are marked in yellow in the paper. The main corrections in the paper and the responds to the reviewer’s comments are as flowing:

  1. Response to comment: Complement genomic content differences.

Response: According to your comments, we have made corresponding supplements in the article. The pictures in the article will be displayed as supplementary materials (Figure S1, Figure S2).

  1. Response to comment: What are the genes which differ.

Response: According to your comments, the specific differential genes have been shown in the Figure 1A, Figure 2A and Table 8.

  1. Response to comment: What appears to be the source of these genes? Are they orthologous to anything?

Response: BmNPV is a baculovirus and has high homology with AcMNPV. In fact, the BmNPV genome is annotated with reference to AcMNPV.

  1. Response to comment: Could they play a role in differences in virulence?

Response: Thank you for your valuable questions and ideas. In fact, the question you asked is exactly the experiment we are currently conducting to verify. However, the number of different genes between the two viruses exceeded our expectations. So I’m very sorry that we still have no way to give you a clear answer. But we have added relevant research in the article. (Line 335-337, page 14).

  1. Response to comment: Line 114: Citation here is missing.

Response: I am very sorry for our negligence, and we have corrected the errors cited parts in the article (Line 143,page 3), (Line 162,page 4) , (Line 181, page 5), (Line 186 and 190, page 5), (Line 221, page 7), (Line 230, 232 and 234, page 7), (Line 254, page 8), (Line 287 and 299, page 9).

  1. Response to comment: Is there any comparison that can eb made to a reference strain from elsewhere?

Response: Thank you for your suggestion. We added the genomic differences between the two viruses and other reference strains in the Table S1. Unfortunately, we are unable to proceed with differences in pathogenicity. All of our silkworms have now reached the fifth instar and are ready to form cocoons. However, the difference in pathogenicity requires the use of second-instar silkworms. In the article, we have provided articles about the comparison of the pathogenicity of different strains of BmNPV for reference (Line 66-73, page 2).

  1. Response to comment: Error in description: concentraTable 21

Response: We are sorry that the picture and the corresponding description have shifted due to our negligence. We have changed the part of Figure 1 (Line 178-179, page 5).

  1. Response to comment: Figure 2: Error in labels of barplots.

Response: We have re-edited to the Figure 2.

  1. Response to comment: Figure 3 & 4: label which genes are orphans, which are shared with the other BmNPV & which are conserved Baculovirus genes.

Response: Considering that there is little effective information in Figure 3 and figure 4, we decide to delete Figure 3 and Figure 4. Accordingly, we mapped the genome linear map of BmNPV ZJ and BmNPV YN to show the genome information of the two viruses. The pictures in the article will be displayed as supplementary materials (Figure S1, Figure S2).

  1. Response to comment: Figure 3 & 4: A ribbon plot connecting contiguous regions of these two genomes & to a BmNPV reference genome would be incredibly helpful for understanding how these genomes differ.

Response: We submit the BmNPV ZJ and BmNPV YN genome line graphs as supplementary materials (Figure S1, Figure S2). If necessary, we will also supplement the BmNPV T3 linear graph.

  1. Response to comment: Line 223: empty parentheses?

Response: I apologize for this error due to our negligence. We have added the correct citation in parentheses in line 234 on page 7 of the article.

  1. Response to comment: Figure 5: Clearer labelling would be appreciated.

Response: We replace the evalue in Table 5 with the mutation type.

  1. Response to comment: Reflect the genome similarity of different reference strains.

Response: Thank you for your comments, this could enrich our article. we compared the nucleotide sequences of the two virus strains with other BmNPV reference strains and revised the article (Line 225-227, page7). The comparison results are shown in the Table S1 in the article.

  1. Response to comment: Line 313: SNPEff citation

Response: We have added relevant references in the method (Line 155, page 4).

We tried our best to improve the manuscript and made some changes in the manuscript.  These changes will not influence the content and framework of the paper. Revised portion are marked in yellow in the paper.

We appreciate for your warm work earnestly, and hope that the correction will meet with approval.

Once again, thank you very much for your comments and suggestions.

Round 2

Reviewer 2 Report

This reviewer has no further comments on the manuscript and suggests careful corrections of English during the proof stage. 

Author Response

Dear Reviewer:

Thank you for your comments concerning our manuscript entitled “Pathogenicity detection and genome analysis of two different geographic strains of BmNPV” (insects-1337816). Those comments are all valuable and very helpful for revising and improving our paper. We have studied comments carefully and have made correction which we hope meet with approval. Revised portion are marked in yellow in the paper. The main corrections in the paper and the responds to the reviewer’s comments are as flowing:

1.Response to comment: Careful corrections of English during the proof stage.

Response: Thank you for your comments. We have revised the whole manuscript carefully and tried to avoid any grammar or syntax error. In addition, we have asked several colleagues who are skilled authors of English language papers to check the English. Revised portion are marked in yellow in the paper.

Other changes:

Line 25-26 and 30-31 page 1, line 118-123 page 3, line 189-194 page 5 and line 336-339 page 14, “BmNPV virus observed by electron microscopic” was added, and marked in red in the paper.

We tried our best to improve the manuscript and made some changes in the manuscript. These changes will not influence the content and framework of the paper. Revised portion are marked in yellow in the paper.

We appreciate for your warm work earnestly, and hope that the correction will meet with approval.

Once again, thank you very much for your comments and suggestions.

Reviewer 3 Report

The authors have improved the language of the manuscript and have clarified some points that were unclear before. While the authors have not taken my suggestions for changing the tables/figures, the manuscript has improved in quality to a presentable level. I still feel the language could be tightened up in a few places, but this can be quickly fixed with another reread.

Author Response

Dear Reviewer:

Thank you for your comments concerning our manuscript entitled “Pathogenicity detection and genome analysis of two different geographic strains of BmNPV” (insects-1337816). Those comments are all valuable and very helpful for revising and improving our paper. We have studied comments carefully and have made correction which we hope meet with approval. Revised portion are marked in yellow in the paper. The main corrections in the paper and the responds to the reviewer’s comments are as flowing:

1.Response to comment: The language could be tightened up in a few places

Response: Thank you for your comments.We have revised the whole manuscript carefully and tried to avoid any grammar or syntax error. In addition, we have asked several colleagues who are skilled authors of English language papers to check the English. Revised portion are marked in yellow in the paper.

Other changes:

Line 25-26 and 30-31 page 1, line 118-123 page 3, line 189-194 page 5 and line 336-339 page 14, “BmNPV virus observed by electron microscopic” was added, and marked in red in the paper.

We tried our best to improve the manuscript and made some changes in the manuscript. These changes will not influence the content and framework of the paper. Revised portion are marked in yellow in the paper.

We appreciate for your warm work earnestly, and hope that the correction will meet with approval.

Once again, thank you very much for your comments and suggestions.
